# Dynamic Nutrition Strategies for Anorexia Nervosa: Marker-Based Integration of Calories and Proteins

**DOI:** 10.3390/nu17030560

**Published:** 2025-01-31

**Authors:** Eugenia Dozio, Martina Alonge, Gianluca Tori, Andrea Caumo, Rina Giuseppa Russo, Edoardo Scuttari, Leonardo Fringuelli, Ileana Terruzzi

**Affiliations:** 1Villa Miralago, Center for the Treatment of Eating Disorders, Cuasso al Monte, 21050 Varese, VA, Italy; eugenia.dozio@villamiralago.it (E.D.); rina.russo@villamiralago.it (R.G.R.); e.scuttari@villamiralago.it (E.S.); leonardo.fringuelli@villamiralago.it (L.F.); 2Biology Applied to the Sciences of Nutrition, Università degli Studi di Milano, 20133 Milano, MI, Italy; martina.alonge@studenti.unimi.it; 3Alkemy S.p.A., 20124 Milano, MI, Italy; gianluca.tori@alkemy.com; 4Department of Biomedical Sciences for Health, Università degli Studi di Milano, 20133 Milano, MI, Italy; andrea.caumo@unimi.it

**Keywords:** anorexia nervosa, nutritional strategies, caloric intake, protein intake, markers, body composition, phase angle, body cell mass, fat-free mass, fat mass, hydration status

## Abstract

Background/Objectives: Anorexia nervosa (AN) is a severe psychiatric disorder characterized by profound nutritional deficits and significant alterations in body composition, cellular integrity, and hydration. Nutritional rehabilitation is critical not only for weight restoration but also for improving body composition and metabolic functions. However, optimal strategies for integrating caloric and protein intake to achieve balanced recovery remain underexplored. This study aims to evaluate the interactions between caloric/protein intake and time on quantitative (weight and BMI) and qualitative (body composition and cellular health) outcomes, and to identify markers that predict recovery trajectories and guide personalized nutritional interventions. Methods: This retrospective observational study analyzed 79 patients with AN admitted to Villa Miralago for six months of nutritional rehabilitation. Anthropometric and body composition parameters—including body weight (BW), body mass index (BMI), fat mass (FM), fat-free mass (FFM), body cell mass (BCM), phase angle (PA), and hydration markers (TBW and ECW)—were assessed at baseline (T0), 3 months (T1), and 6 months (T2). Generalized Estimating Equations (GEEs) were used to evaluate the effects of caloric and protein intake over time. Results: Significant increases in BW (+6.54 kg, *p* < 0.0001) and BMI (+2.47 kg/m^2^, *p* < 0.0001) were observed, alongside improvements in FM, FFM, and BCM. PA increased significantly (+0.47°, *p* < 0.0001), indicating enhanced cellular health. TBW increased (+1.58 L, *p* < 0.0001), while ECW% decreased, reflecting improved fluid distribution. Caloric intake predominantly influenced early fat mass recovery, while protein intake was crucial for preserving lean tissues and promoting cellular regeneration. Interaction effects between caloric/protein intake and time revealed dynamic changes in body composition, underscoring the need for adaptive strategies. Conclusions: This study highlights the importance of a dynamic, marker-based approach to nutritional rehabilitation in AN. Integrating caloric and protein intake with advanced body composition and hydration markers enables personalized interventions and balanced recovery, shifting AN treatment toward a focus on qualitative improvements overweight restoration alone.

## 1. Introduction

Anorexia nervosa (AN) is a severe and complex psychiatric disorder marked by extreme caloric restriction, significant weight loss, and profound nutritional deficits [1]. These deficits result in a cascade of physiological dysfunctions, including alterations in body composition, metabolic health, and cellular integrity, which contribute to the significant morbidity and mortality associated with the disorder [2,3,4]. The restoration of weight and body composition, alongside the improvement of metabolic and cellular function, is a cornerstone of treatment [5,6,7,8]. However, the optimal nutritional strategies to achieve these outcomes remain an area of active investigation. Nutritional rehabilitation is the primary intervention for AN, aiming not only to restore body weight but also to improve qualitative aspects of recovery, such as body composition, cellular function, and fluid balance [1,6]. While caloric intake is essential to reverse weight loss [9], protein intake plays a critical role in supporting lean body mass recovery and cellular repair [10]. Despite this, the precise interplay between caloric and protein intake, time, and individual patient responses remains insufficiently researched and understood [11]. Despite the primary focus in the nutritional rehabilitation of anorexic patients being the determination of appropriate energy intake and the types of foods and macronutrients [6,11], the concept of a dynamic nutrition approach [12] has rarely been considered. Such an approach would allow for nutritional adaptations during the rehabilitation process, ensuring optimal adjustments in energy and protein intake to support both metabolic recovery and the reconstruction of body composition, particularly with respect to protein restoration and lean body mass development. This gap in knowledge underscores the need for individualized approaches to optimize nutritional rehabilitation. In this context, identifying reliable markers that can predict recovery trajectories and inform clinical decision-making would be of significant value and support. Markers derived from body composition parameters—such as (BW) body weight; (BMI) body mass index; (FM) fat mass; (FM%) fat mass percentage; (FFM) fat-free mass; (FFM%) fat-free mass percentage; (BMC) body cell mass; (BMCI) body cell mass index; (PA) phase angle; (TBW) total body water; (TBW%) total body water percentage; (ECW) extracellular water; (ECW%) extracellular water percentage—offer valuable insights into the qualitative progress of recovery [13]. These markers not only provide a deeper understanding of the physiological changes during nutritional rehabilitation but may also serve as indicators to refine and personalize interventions, enhancing their efficacy [14,15].

This study seeks to evaluate the impact of caloric and protein intake on a comprehensive set of anthropometric and body composition parameters in a cohort of patients with AN over a six-month rehabilitation period. By analyzing the dynamic interactions between dietary intake, time, and recovery outcomes, we aim to identify key markers that can serve as indicators of effective nutritional interventions. Specifically, these markers could provide actionable insights to optimize caloric and protein intake for individual patients, monitor qualitative improvements in cellular and metabolic function, predict recovery trajectories, and refine rehabilitation strategies. The novelty of this study lies in its focus on integrating traditional anthropometric measures, such as weight and BMI, with advanced body composition metrics (e.g., BCM and PA) to establish a framework for individualized nutritional rehabilitation. By identifying markers of recovery, we aim to bridge the gap between clinical outcomes and mechanistic insights, ultimately improving the effectiveness of interventions for patients with AN. We hypothesize that specific body composition parameters and fluid balance serve as reliable markers of recovery in patients undergoing nutritional rehabilitation for AN. Our objectives are as follows:To assess the interactions between caloric/protein intake and time on both quantitative (weight and BMI) and qualitative (BCM, PA, and FM%) outcomes.To identify markers that predict recovery trajectories and guide personalized nutritional interventions.To provide evidence-based recommendations to optimize rehabilitation strategies and improve clinical outcomes.

## 2. Materials and Methods

### 2.1. Study Design

This retrospective observational study was conducted at the Villa Miralago (VM), Center for the treatment of Eating Disorders (Cuasso al Monte, VA, Italy). The total observation period was six months, examining the medical records of 79 patients admitted between January 2018 and January 2024. Inclusion criteria included a diagnosis of anorexia nervosa (AN) according to DSM-V [16] in subjects of both sexes, Caucasian race, aged 16 years or older, intensive nutritional rehabilitation treatment, and a minimum hospitalization period of six months. Exclusion criteria included patients under 16 years of age, diagnoses other than those specified (e.g., eating disorders not otherwise specified, EDNOS), and hospitalizations shorter than six months. Furthermore, patients who met the inclusion criteria but had clinical or psychopathological conditions preventing adequate monitoring of the study parameters were excluded from the study. These included severe, non-stabilized psychiatric comorbidities or internal medical conditions incompatible with the nutritional rehabilitation protocol, which are also criteria for non-acceptance into the program at Villa Miralago.

### 2.2. Procedure

#### 2.2.1. Initial Assessment at Time 0 (T0)

At time 0 (T0), corresponding to the patients’ admission to Villa Miralago, each patient was welcomed by the multidisciplinary team (psychiatrists, internists, nutrition specialists, nurses, psychologists, psychotherapists, educators, dietitians, art therapists, and kinesiologists) and underwent a comprehensive evaluation, including a Nutritional Assessment.

In agreement with the reference team, the internist prescribes the necessary blood tests to monitor the rehabilitation of the patients, with particular attention to the early stages of the process, when their conditions are more severe. The blood markers used to assess the severity of malnutrition and to monitor nutritional and metabolic status include the following: hemoglobin, hematocrit, leukocytes, total lymphocytes, transferrin, blood glucose, creatinine, vitamin B12, folate, ferritin, glucose, creatinine, ALT and AST transaminases, sodium, potassium, chloride, magnesium, calcium, inorganic phosphate, total proteins, albumin, vitamin D, and zinc. These tests are repeated during the patient’s hospitalization whenever deemed necessary to reassess their clinical condition and monitor the relevant parameters. Blood samples are sent to the hospital laboratory (Varese, Italy) for analysis.

Upon admission and throughout the hospitalization period, patients underwent routine psychological and psychiatric assessments, followed by highly individualized therapeutic interventions for the management of psychiatric comorbidities and pathological addictions. As part of these assessments, standardized psychometric tests were administered, including the EDI-3 (Eating Disorder Inventory-3) to analyze psychological characteristics associated with eating disorders, the BUT (Body Uneasiness Test) to assess body discomfort and dysmorphophobia, and the MMPI-2 (Minnesota Multiphasic Personality Inventory-2) to explore personality profiles and potential comorbid psychopathologies. These tools allow for a more precise clinical evaluation and guide the planning of therapeutic interventions.

The personalized pharmacological treatment prescribed following the initial psychiatric evaluation was subsequently monitored and adjusted according to the evolution of the patient’s clinical and psychological condition throughout the hospitalization period.

The nutritionist conducted a detailed nutritional anamnesis, which explored the patient’s physiological and weight history, including birth weight, minimum and maximum weight achieved (with an indication of age), and significant events related to the onset of the disorder. Further aspects considered were aggravating and maintaining factors, such as an analysis of dietary practices and past eating habits, as well as daily habits, including the frequency and intensity of physical activity, the presence of “fear foods”, and weight control behaviors.

Based on the collected information, a personalized nutritional plan was developed. This plan was periodically adjusted during the hospitalization to account for weight changes and psychophysical progress. Each adjustment was discussed and shared with the patient to foster a strong therapeutic alliance.

The adopted nutritional plan included the distribution of meals into four main meals (breakfast, lunch, afternoon snack, and dinner), with the possible addition of a mid-morning snack in certain cases. Caloric and protein intake was personalized, ranging from 24.81 to 63.64 kcal/kg/day and 1.54 to 2.87 g/kg/day of protein within the patient cohort included in the study.

Caloric and protein intake adjustments were not necessarily implemented according to the standard timing suggested by guidelines but were instead tailored based on the multidisciplinary team’s evaluation. This approach allowed for the definition of a personalized therapeutic plan that considered multiple factors, including nutritional impairment severity, the patient’s ability to tolerate caloric and protein increases, the presence of comorbidities, and psychological or behavioral factors that might have hindered nutritional recovery. The multidisciplinary team’s involvement ensured a dynamic modulation of nutrient intake, supporting a gradual and individualized approach. This approach not only addressed the clinical and metabolic needs of the patient but also translated into structured dietary choices. To this end, three menu options were designed based on daily caloric intake: Menu A (~1400 kcal/day): Protein: 20% of total calories; Lipids: 34% of total calories, with saturated fats at 10%; Carbohydrates: 46% of total calories; Fiber: 19 g/day. Menu B (~1700 kcal/day): Protein: 18% of total calories; Lipids: 29% of total calories, with saturated fats at 8%; Carbohydrates: 53% of total calories; Fiber: 22 g/day. Menu C (~2000 kcal/day): Protein: 17% of total calories; Lipids: 25% of total calories, with saturated fats at 7%; Carbohydrates: 58% of total calories; Fiber: 25 g/day.

#### 2.2.2. Longitudinal Monitoring of Anthropometric and Body Composition Parameters

At baseline (T0), as well as at 3 months (T1) and 6 months (T2) post-admission, anthropometric and body composition parameters were recorded for each patient. Body weight was measured using an electronic scale (Kern MPD-E, Arroweld Group Italy, Zanè, VI, Italy; capacity: 250 kg; resolution: 0.1 kg), while height was determined with a stadiometer (Wunder W030299—Trezzo sull’Adda, MI, Italy). Body composition was assessed through vector bioelectrical impedance analysis (BIVA) using the Akern BIA101 device, which operates at 250 μA and 50 kHz mono-frequency.

### 2.3. Bioelectrical Impedance Vector Analysis

Whole-body vector bioelectrical impedance analysis (BIVA) was performed using a 50 kHz phase-sensitive impedance analyzer (Akern BIA101, Pontassieve, FI, Italy). Adhesive inductive electrodes with a current of 800 mA were applied to the right hand and right foot dorsally, corresponding to the metacarpal epiphysis of the third finger and the metatarsal epiphysis of the second toe; the sensing electrodes are positioned between the distal prominences of the radius and ulna and between the medial and lateral malleolus of the ankle [17]. The device recorded resistance (Rz) and reactance (Xc) values, which were used to compute the phase angle (PA). The interpretation of BIVA utilized standardized Rz and Xc values plotted on resistance–reactance graphs based on the methodology previously described [18]. This approach enabled direct evaluation of body composition without reliance on predictive models or body weight. The analyzed bioelectrical parameters included: fat mass (FM), percentage of fat mass (FM%), fat-free mass (FFM), percentage of fat-free mass (FFM%), Body cell mass (BCM), Body cell mass index (BCMI), Phase angle (PA), total body water (TBW), percentage of total body water (TBW%), extracellular water (ECW), and percentage of extracellular water (ECW%).

### 2.4. Statistical Analysis

This retrospective observational study aimed to investigate the relationship between kilocalorie (kcal) and protein intake and the improvement of physiological parameters in individuals with eating disorders. A statistical modeling approach was adopted to analyze the dynamics of nutritional intake and temporal progression. The dependent variables in the analysis were the physiological parameters (e.g., BMI and BMCI), while the independent variables included the quantity of kcal or protein consumed, time (expressed as T0, T1, T2, corresponding to different temporal points), and the interaction between nutritional intake and time. The primary hypothesis proposed that the interaction between nutrient intake and time significantly influences improvements in physiological health metrics in patients with eating disorders. To address these objectives, Generalized Estimating Equations (GEE) were employed as the modeling technique. GEE is particularly suitable for longitudinal data analysis as it allows the incorporation of an autoregressive correlation structure within individual patients over time [19]. This method effectively manages repeated measures, where observations within the same subject are not independent [20]. The statistical analyses were conducted using the R software environment, version 4.3.1. Specifically, the “geeglm” function from the “geepack” package was used to fit the GEE models. This function enables parameter estimation while accounting for the specified correlation structure among repeated measures. Regression coefficients from the models were tested using Wald tests to determine their statistical significance. This analysis underscores the utility of GEE in examining complex interactions over time within longitudinal nutritional data. The methodological framework presented here provides a robust foundation for further studies exploring the temporal effects of dietary interventions on physiological outcomes in clinical settings.

## 3. Results

### 3.1. Longitudinal Changes in Body Composition in AN Patients

The longitudinal analysis of data collected from 79 patients with anorexia nervosa (AN) undergoing a nutritional rehabilitation program revealed significant changes in anthropometric and body composition parameters, measured at the start of observations (T0), after 3 months (T1), and 6 months (T2). These findings align with prior studies highlighting the effectiveness of structured nutritional interventions in improving body composition and restoring metabolic balance in patients with AN [6].

BW (Figure 1A) significantly increased (*p* = 3.01592 × 10^−19^), starting from a baseline mean value of 38.75 + 6.36 kg to 45.29 + 6.54 kg at 6 months (Table 1). This increase was accompanied by a significantly (*p* = 4.85064 × 10^−19^) linear rise in BMI (Figure 1B), which consistently increased from 14.52 + 2.12 Kg at baseline to 16.99 + 1.95 kg at 6 months, reflecting the progressive weight recovery in line with the adopted nutritional plan. FM (Figure 1C) significantly increased (*p* = 7.4 × 10^−11^), rising from 3.17 + 2.20 kg to 5.43 + 3.46 kg over the observation period (Table 1). FM% (Figure 1D) also showed a progressive increase (*p* = 1.93 × 10^−8^) rising from 7.83% + 4.09 to 11.37% + 6.03. These trends are consistent with the expected fat mass recovery in AN patients during initial nutritional rehabilitation. FFM (Figure 1E) increased steadily (*p* = 2.59 × 10^−8^) starting at 35.53 + 5.03 kg and reaching 39.86 + 4.25 kg at 6 months (Table 1). In contrast to the absolute increase in FFM, FFM% (Figure 1F) significantly decreased (*p* = 2.59 × 10^−8^) from over 92.17% + 4.09 at baseline to 88.63% + 6.03 at 6 months (Table 1), reflecting the proportional increase in fat mass. BCM (Figure 1G) and the body cell mass index (BCMI) (Figure 1H) showed a consistent increase (*p* = 4.52918 × 10^−14^; *p* = 3.83 × 10^−14^, respectively) (Table 1), suggesting an improvement in body tissue quality and a recovery of metabolically active components. PA (Figure 1I), an indicator of cell quality and membrane integrity, significantly increased (*p* = 4.95 × 10^−7^) from approximately 4.73° + 1.04 to 5.20° + 0.87, indicating an improvement in the nutritional and functional status of cellular membranes. TBW (Figure 1L) showed a linear and significant increase (*p* = 1.33 × 10^−8^), rising from 28.05 + 3.29 L at baseline to 29.63 + 3.16 L at 6 months. This increase aligns with the recovery of fat-free mass and the overall improvement in nutritional status. Conversely, the percentage of total body water (TBW%) (Figure 1M) relative to body weight progressively decreased (*p* = 6.68 × 10^−14^) from 73.46 + 8.58% to 65.96 + 5.55%, consistent with the increase in fat mass and overall body weight. These changes in hydration markers reflect a normalization of fluid distribution during recovery. The absolute extracellular water (ECW) showed a non-linear trend (Figure 1N), with an initial decrease from 14.77 + 2.39 L to 14.67 + 1.88 L at 3 months (*p* = 0.593261), followed by a slight increase to 14.72 + 1.89 L at 6 months (*p* = 0.612387). These results indicate a temporary redistribution of extracellular fluids, likely associated with metabolic changes and the gradual restoration of body composition. The percentage of extracellular water (ECW%) showed a progressive decline (Figure 1O), decreasing significantly (*p* = 0.007383) from initial values of 49.88 + 14.78% to 47.56 + 12.03% at 6 months. This finding suggests a positive shift toward intracellular fluid compartmentalization, associated with metabolic recovery and cellular health.

### 3.2. Impact of Caloric Intake, Time, and Their Interaction on Body Composition in AN Patients

Caloric intake (kcal/kg/day) showed significant associations with several body composition parameters, although the effects varied depending on the parameter and its interaction with time.

The association between kcal/kg/day and BW (Table 2A) is reflected in an estimate of −0.1965 −0.1965 (Std. Err. = 0.0669; Wald = 8.6381; *p* = 0.0033), showing a statistically significant negative association. This suggests that an increase in kcal/kg/die is associated with a slight but significant decrease in BW. Time alone shows a positive but non-significant effect (Estimate = 0.6991; Std. Err. = 0.5424; Wald = 1.6612; *p* = 0.1974). This implies that time alone does not have a significant impact on BW. The interaction term (kcal/kg/day × time) has an estimate of 0.0094 (Std. Err. = 0.01301; Wald = 0.5214; *p* = 0.4703). This result suggests that the interaction between kcal/kg/die and time does not significantly influence BW. For BMI (Table 2B), kcal/kg/day shows a significant negative effect (Estimate = −0.0815; Std. Err. = 0.0255; Wald = 10.2371; *p* = 0.0014), suggesting that an increase in kcal/kg/die is associated with a significant decrease in BMI. Time alone has a non-significant positive effect on BMI (Estimate = 0.1777; Std. Err. = 0.2007; Wald = 0.7840; *p* = 0.3759), suggesting that time alone does not significantly influence BMI. The interaction term is also non-significant (Estimate = 0.0057; Std. Err. = 0.0048; Wald = 1.4267; *p* = 0.2323), indicating that the interaction does not have a statistically significant effect on BMI. In the analysis of FM (Table 2C), kcal/kg/day demonstrates a negative but non-significant association (Estimate = −0.0368; Std. Err. = 0.0251; Wald = 2.1490; *p* = 0.1427). Time shows a significant positive effect (1.2983; Std. Err. = 0.2506; Wald = 26.8402; *p* < 0.0001). The interaction term (Estimate = −0.0240; Std. Err. = 0.0059; Wald = 16.5107; *p* < 0.0001) indicates a significant moderating effect, with kcal/kg/day over time associated with a decrease in FM. The results for FM% (Table 2D) reveal a non-significant negative association with kcal/kg/day (Estimate = −0.0744; Std. Err. = 0.0468; Wald = 2.5254; *p* = 0.1120), suggesting that changes in kcal/kg/die alone do not significantly influence FM%. Time shows a significant positive association (Estimate = 2.2488; Std. Err. = 0.4628; Wald = 23.6057; *p* < 0.0001). This finding suggests that, as time progresses, FM significantly increases. The interaction term (Estimate = −0.0434; Std. Err. = 0.0111; Wald = 15.1738; *p* = 0.0001) is significant, indicating that an increase in kcal/kg/die over time is associated with a decrease in FM%. For FFM (Table 2E), kcal/kg/day shows a significant negative association (Estimate = −0.1778; Std. Err. = 0.0531; Wald = 11.2113; *p* = 0.0008), suggesting that, independently, an increase in kcal/kg/die is associated with a significative slight decrease in FFM. Time alone has a non-significant negative effect (Estimate = −0.5159; Std. Err. = 0.4545; Wald = 1.2886; *p* = 0.2563). This indicates that the passage of time, by itself, does not significantly influence FFM. The interaction term (Estimate = 0.0314; Std. Err. = 0.0130; Wald = 7.8052; *p* = 0.0052) indicates a significant positive influence, suggesting that the combined effect of kcal/kg/die intake over time positively influences FFM. In the context of FFM% (Table 2F), kcal/kg/day has a non-significant positive effect (Estimate = 0.0744; Std. Err. = 0.0468; Wald = 2.525; *p* = 0.1120). Time shows a significant negative association (Estimate = −2.2488; Std. Err. = 0.4628; Wald = 23.6057; *p* = 0.0001), suggesting that, as time progresses, FFM% significantly decreases. The interaction term (Estimate = 0.0433; Std. Err. = 0.0111; Wald = 15.1738; *p* = 0.0001) suggests that kcal/kg/die and time together have a significant moderating effect on FFM%, with an increase in kcal/kg/die over time being associated with an increase in FFM%. The findings for BCM (Table 2G) show a significant negative association with kcal/kg/day (Estimate = −0.1397; Std. Err. = 0.0410; Wald = 11.5918; *p* = 0.0007): an increase in kcal/kg/die is associated with a significant decrease in BCM. Time alone has a non-significant negative effect on BCM (Estimate = −0.3857; Std. Err. = 0.3067; Wald = 1.5817; *p* = 0.2085). The interaction term (Estimate = 0.0231; Std. Err. = 0.0077; Wald = 8.9786; *p* = 0.0027) indicates that the interaction between kcal/kg/die and time has a significant positive effect on BCM. The data suggest that the impact of kcal/kg/die on BCM becomes less negative or potentially positive over time. In Table 2H, the relationships between BCMI and kcal/kg/day (Estimate = −0.0521; Std. Err. = 0.0152; Wald = 11.6777; *p* = 0.0006) and BCMI and time alone (Estimate = −0.2219; Std. Err. = 0.1122; Wald = 3.9074; *p* = 0.0481) show a significant negative association. This suggests that both caloric intake and time alone are associated with a significant decrease in BCMI. The interaction between kcal/kg/die and time (Estimate = 0.0107; Std. Err. = 0.0028; Wald = 14.2799; *p* = 0.0002) is significant, showing that the interaction has a significant moderating effect on BCMI, with an increase in kcal/kg/die over time being associated with an increase in BCMI. The results in Table 2I concerning PA reveal that kcal/kg/day has a significant negative association (Estimate = −0.0245; Std. Err. = 0.0107; Wald = 5.2481; *p* = 0.0220), suggesting that an increase in kcal/kg/die is significantly associated with a decrease in PA. Time alone has a non-significant negative effect on PA (Estimate = −0.0926; Std. Err. = 0.0965; Wald = 0.9193; *p* = 0.3377). The interaction term for kcal/kg/die and time (Estimate = 0.0044; Std. Err. = 0.0025; Wald = 3.0758; *p* = 0.0795) does not have a significant effect on PA. As reported for TBW in Table 2L, kcal/kg/day shows a non-significant negative effect (Estimate = −0.0069; Std. Err. = 0.0262; Wald = 0.0700; *p* = 0.7913). Time shows a marginally significant positive association (Estimate = 0.5205; Std. Err. = 0.2796; Wald = 3.4659; *p* = 0.0626). The interaction term (Estimate = −0.0067; Std. Err. = 0.0067; Wald = 0.9590; *p* = 0.3274) is non-significant. Examining TBW% (Table 2M), a significant positive association with kcal/kg/day is evident (Estimate = 0.4527; Std. Err. = 0.0994; Wald = 20.7402; *p* < 0.0001). This finding suggests that an increase in kcal/kg/die is significantly associated with an increase in TBW%. Time also shows a significant positive effect (Estimate = 1.4780; Std. Err. = 0.7523; Wald = 3.8596; *p* = 0.0495), suggesting that, as time progresses, TBW% increases significantly. The interaction term (Estimate = −0.0691; Std. Err. = 0.0197; Wald = 12.3327; *p* = 0.0005) indicates a significant negative effect, with an increase in kcal/kg/die over time being associated with a decrease in TBW%. The data for ECW (Table 2N) highlight a significant positive relationship with kcal/kg/day (Estimate = 0.0477; Std. Err. = 0.0233; Wald = 4.1900; *p* = 0.0407): an increase in kcal/kg/die is significantly associated with an increase in ECW. Time shows a significant positive association (Estimate = 0.5790; Std. Err. = 0.2440; Wald = 5.6303; *p* = 0.0177), suggesting that, as time progresses, ECW increases significantly. The interaction term (Estimate = −0.0150; Std. Err. = 0.0063; Wald = 5.6871; *p* = 0.0171) is significant, indicating a decrease in ECW with an increase in kcal/kg/day over time. From the analysis of ECW% (Table 2O), kcal/kg/day shows a non-significant positive association (Estimate = 0.1170; Std. Err. = 0.0785; Wald = 2.2238; *p* = 0.1359). Time has a non-significant positive effect (Estimate = 0.8302; Std. Err. = 0.7743; Wald = 1.1498; *p* = 0.2836). The interaction term for kcal/kg/die and time (Estimate = −0.0311; Std. Err. = 0.0217; Wald = 2.0581; *p* = 0.1514) shows that the interaction does not significantly affect ECW%. These data suggest that neither caloric intake and time alone nor their interaction significantly influence ECW%.

### 3.3. Impact of Protein Intake, Time, and Their Interaction on Body Composition in AN Patients

Protein intake (g/kg/day) demonstrated distinct effects compared to caloric intake, underscoring its critical role in supporting lean body mass and metabolic recovery.

The effect of P g/kg/day on BW (Table 3A) is estimated at −28.3010 (Std. Err. = 6.3084; Wald = 20.1263; *p* < 0.0001), showing a statistically significant negative association, with an increase in P g/kg/die being significantly associated with a decrease in BW. Time alone has a significant negative effect (Estimate = −5.1566; Std. Err. = 1.2878; Wald = 16.0336; *p* = 0.0001) indicating that, as time progresses, BW decreases significantly. The interaction term (*p* g/kg/day × time) has an estimate of 3.8241 (Std. Err. = 1.4618; Wald = 6.8435; *p* = 0.0089), indicating a significant moderating effect on BW, with an increase in P g/kg/die over time being associated with an increase in BW. The relationship between P g/kg/day and BMI shown in Table 3B reveals a significant negative effect (Estimate = −10.0500; Std. Err. = 2.0832; Wald = 23.2738; *p* < 0.0001), suggesting that an increase in P g/kg/die is significantly associated with a decrease in BMI. Time alone has a significant negative effect (Estimate = −1.9178; Std. Err. = 0.4453; Wald = 18.5469; *p* < 0.0001), indicating that, as time progresses, BMI decreases significantly. The interaction term for P g/kg/die and time is significant (Estimate = 1.4456; Std. Err. = 0.5085; Wald = 8.0822; *p* = 0.0045). This result suggests that the interaction between P g/kg/die and time has a significant moderating effect on BMI, with an increase in P g/kg/die over time being associated with an increase in BMI. The findings for FM (Table 3C) show a significant negative association with P g/kg/day (Estimate = −33.4522; Std. Err. = 10.2528; Wald = 10.6456; *p* = 0.0011), suggesting that an increase in protein intake is significantly associated with a decrease in FM. The significant negative effect of time alone on FM (Estimate = −4.0601; Std. Err. = 1.2785; Wald = 10.0854; *p* = 0.0015) suggests that, as time progresses, FM decreases significantly. The interaction term (Estimate = 3.0709; Std. Err. = 1.5625; Wald = 3.8628; *p* = 0.0494) indicates a significant moderating effect on FM, with an increase in P g/kg/die over time being associated with an increase in FM. In the case of FM% (Table 3D), P g/kg/day shows a non-significant negative effect (Estimate = −17.2348; Std. Err. = 9.9717; Wald = 2.9872; *p* = 0.0839). Time alone shows a negligible non-significant effect (Estimate = −0.0111; Std. Err. = 1.1361; Wald = 0.0001; *p* = 0.9922). The interaction term (Estimate = −0.6905; Std. Err. = 1.4186; Wald = 0.2369; *p* = 0.6265) is also non-significant. These results suggest that neither protein intake and time alone nor their interaction significantly affect FM%. Table 3E shows a non-significant positive association between FFM and P g/kg/day (Estimate = 2.9994; Std. Err. = 4.9697; Wald = 0.3643; *p* = 0.5461). Time alone has a non-significant negative effect (Estimate = −1.0567; Std. Err. = 0.7152; Wald = 2.1828; *p* = 0.1396). The interaction term (Estimate = 0.7534; Std. Err. = 0.7518; Wald = 1.0041; *p* = 0.3163) is also non-significant. This result suggests that protein intake alone, time alone, and their interaction do not significantly affect FFM. When examining FFM% (Table 3F), P g/kg/day shows a non-significant positive effect (Estimate = 17.2348; Std. Err. = 9.9717; Wald = 2.9872; *p* = 0.0839). Time alone has a negligible non-significant effect (Estimate = 0.0111; Std. Err. = 1.1361; Wald = 0.0001; *p* = 0.9922). The interaction term (Estimate = 0.6905; Std. Err. = 1.4186; Wald = 0.2369; *p* = 0.6265) is also non-significant. Even in the case of FFM%, proteins and time individually or in combination fail to induce any significant effect. Similarly, no effect of protein, time, or their interaction (kcal/kg/day × time) on BCM was observed (Table 3G): P g/kg/day shows a non-significant negative effect (Estimate = −2.6712; Std. Err. = 4.0408; Wald = 0.4370; *p* = 0.5086); time alone has a non-significant negative effect (Estimate = −0.2763; Std. Err. = 0.3362; Wald = 0.6753; *p* = 0.4112); the interaction term (Estimate = 0.1409; Std. Err. = 0.3365; Wald = 0.1754; *p* = 0.6754) is also non-significant. Overall, neither protein intake (Estimate = −1.4882; Std. Err. = 1.3839; Wald = 1.1563; *p* = 0.2822), time (Estimate = −0.0905; Std. Err. = 0.1282; Wald = 0.4983; *p* = 0.4803), nor their interaction (Estimate = 0.0582; Std. Err. = 0.1298; Wald = 0.2010; *p* = 0.6539) significantly influence BCMI (Table 3H). In the context of PA (Table 3I), P g/kg/day shows a non-significant negative effect (Estimate = −0.7505; Std. Err. = 0.7923; Wald = 0.8972; *p* = 0.3435). Time alone has a non-significant positive effect (Estimate = 0.1173; Std. Err. = 0.0925; Wald = 1.6102; *p* = 0.2045). The interaction term (Estimate = −0.1264; Std. Err. = 0.0988; Wald = 1.6361; *p* = 0.2009) is also non-significant. Water balance also appears to be unaffected by the isolated or combined effects of protein intake and time. For TBW (Table 3L), P g/kg/day shows a non-significant positive effect (Estimate = 1.2879; Std. Err. = 3.6085; Wald = 0.1274; *p* = 0.7212). Time alone has a non-significant negative effect (Estimate = −1.5337; Std. Err. = 1.0626; Wald = 2.0835; *p* = 0.1489). The interaction term (Estimate = 1.3217; Std. Err. = 1.0620; Wald = 1.5488; *p* = 0.2133) is also non-significant. Analysis for TBW% (Table 3M) confirms that P g/kg/day (Estimate = 12.9758; Std. Err. = 7.5653; Wald = 2.9418; *p* = 0.0863), time alone (Estimate = −0.5539; Std. Err. = 0.9274; Wald = 0.3567; *p* = 0.5503), and their interaction (Estimate = 1.0637; Std. Err. = 1.1026; Wald = 0.9307; *p* = 0.3347) show non-significant effects. For ECW (Table 3N), P g/kg/day shows a non-significant positive effect (Estimate = 1.0656; Std. Err. = 2.3143; Wald = 0.2119; *p* = 0.6452). Time alone has a non-significant negative effect (Estimate = −0.8824; Std. Err. = 0.5636; Wald = 2.4511; *p* = 0.1174). The interaction term (Estimate = 0.8135; Std. Err. = 0.5909; Wald = 1.8947; *p* = 0.1687) is also non-significant. Finally, protein intake (Estimate = 3.0020; Std. Err. = 3.0696; Wald = 0.9564; *p* = 0.3281), time (Estimate = −0.5166; Std. Err. = 0.4095; Wald = 1.5915; *p* = 0.2071), and their interaction (Estimate = 0.5363; Std. Err. = 0.4391; Wald = 1.4913; *p* = 0.2220) do not have a significant effect on ECW% (Table 3O).

## 4. Discussion

The evaluation of patients with anorexia nervosa (AN) undergoing a six-month nutritional rehabilitation program provided detailed insights into weight recovery and its implications for body composition and nutritional status. The results demonstrate how the dynamic personalized integration of caloric and protein intake, monitored through specific markers, represents a fundamental approach to ensuring balanced, sustainable, and qualitatively significant recovery.

### 4.1. Trends in Body Composition and Hydration During Nutritional Rehabilitation in AN

The progressive increase in body weight (BW) and body mass index (BMI) observed in our study during the six-month rehabilitation period provides clear evidence of the effectiveness of the nutritional plan adopted in promoting weight restoration [1,21]. However, while these parameters are essential for assessing overall progress, our findings indicate that they do not fully capture the quality of recovery [22]. The observed increase in fat mass (FM), both in absolute and relative terms (FM%), reflects the restoration of energy reserves, a primary goal of nutritional rehabilitation, as observed in other studies [1,21,23]. At the same time, the increase in fat-free mass (FFM) in our cohort, in accord with previously data [1,21,23], suggests support for the regeneration of metabolically active tissues. These data align with prior findings indicating that fat-free mass recovery is often coupled with proportional fat mass gains during nutritional rehabilitation [24,25,26]. Nonetheless, the reduction in the relative percentage of FFM compared to total body weight points to an initially imbalanced recovery favoring fat mass, highlighting the importance of carefully monitoring body composition distribution to prevent excessive fat accumulation, which could have negative metabolic or psychological implications in AN patients [20,27,28]. Our data further demonstrate that markers of cellular and functional recovery, such as body cell mass (BCM), body cell mass index (BCMI), and phase angle (PA), provide a unique perspective on qualitative improvement during rehabilitation. These markers have been previously identified as critical indicators of lean body mass recovery and overall nutritional progress in malnourished patients [29]. Consistent increases in BCM and BCMI observed in our study indicate the gradual recovery of metabolically active tissues, a fundamental component for long-term functional recovery. Similarly, the progressive growth in PA recorded in our patients reflects improvements in membrane integrity and overall cell quality. Phase angle improvements have been strongly associated with better clinical outcomes and cellular functionality in malnourished populations [1,21,23]. This parameter emerges as particularly relevant in clinical practice, as our findings confirm its ability to provide insights into cellular health and the functional status of the body that cannot be captured by traditional anthropometric measures, such as BMI or body weight. Hydration markers also played a crucial role in interpreting the physiological responses observed during rehabilitation. The significant increase in total body water (TBW) in our patients aligns with the recovery of FFM, suggesting improved intracellular hydration and overall nutritional status [30,31,32]. Conversely, the reduction in TBW percentage relative to total body weight reflects the increase in FM reserves, indicating a normalization of fluid distribution. Additionally, the behavior of extracellular water (ECW) and its relative percentage (ECW%) in our cohort showed a temporary redistribution of fluids, with a trend toward greater intracellular compartmentalization, signaling improvements in cellular health and metabolic recovery. These findings underscore the importance of integrating hydration markers with body composition parameters in assessing the success of nutritional rehabilitation interventions, as documented in other studies [33].

### 4.2. Interaction Between Caloric and Protein Intake and Recovery Dynamics

Weight recovery in patients with AN, as highlighted by our data, is not a linear or uniform process but rather a dynamic phenomenon characterized by evolving metabolic adaptations over time [34]. Our analysis shows that the early stages of rehabilitation are dominated by a hypermetabolic response, during which the body tends to prioritize restoring energy reserves in the form of fat mass (FM), as evidenced by the proportional increase in FM relative to fat-free mass (FFM), as confirmed in previous studies [24,25,26]. This increase is accompanied by a reduction in the relative percentage of FFM compared to total body weight (FFM%), suggesting that caloric intake, while indispensable at this stage, initially meets immediate survival needs, promoting imbalanced recovery favoring lipid reserves. This result reflects a characteristic biological response in malnourished patients, who require a positive energy balance to stabilize clinical conditions and support vital functions [35]. Simultaneously, our data indicate that protein intake plays a crucial role in preserving FFM and promoting cellular and functional recovery. In patients with AN, adequate protein intake is particularly important to counteract muscle catabolism and support anabolism [8]. However, although weight gain (BW) is not significantly influenced by protein supplementation during the early phases, our results suggest that proteins are utilized by the body to repair and preserve metabolically active tissues rather than contributing to FM accumulation. This observation underscores the importance of timing and proportionality between calories and protein to achieve optimal recovery. In the medium-to-long term, our results show that the interaction between caloric and protein intake results in the progressive stabilization of BW and body mass index (BMI). This balanced recovery, with proportional increases in FM and FFM, reflects a transition towards a qualitative improvement in body composition. The proportionality between these components is essential to prevent metabolic complications, such as excessive FM accumulation, and to reduce the risk of psychological repercussions related to body image perception [20,28].

### 4.3. Body Composition Markers and Recovery Quality

The body composition markers analyzed in our study, such as FM, FFM, body cell mass (BCM), and body cell mass index (BCMI), provided a more detailed and qualitative evaluation of recovery compared to traditional anthropometric parameters. The increases observed in BCM and BCMI during the six months of rehabilitation reflect the recovery of metabolically active tissues, a crucial element for maintaining vital functions and achieving stable nutritional status [22]. These markers indicate that weight recovery is not limited to weight gain but also includes the regeneration of critical cellular components, signaling long-term functional improvements. Additionally, our results show a significant improvement in phase angle (PA) during the rehabilitation period. This parameter, closely related to cellular quality and membrane integrity, suggests that the dynamic integration of caloric and protein intake not only supports tissue regeneration but also contributes to enhanced cellular functions. PA thus emerges as a useful marker for monitoring progress towards functional recovery, beyond simple weight gain.

### 4.4. Role of Hydration Markers in Monitoring Recovery

The hydration markers analyzed in our study also provided essential information about recovery dynamics. The increase in total body water (TBW) was associated with improved intracellular hydration, a positive signal of restored cellular functionality [30,31,32]. At the same time, our data showed a progressive reduction in the percentage of extracellular water (ECW%) relative to TBW, suggesting the normalization of fluid distribution and greater intracellular compartmentalization. This process, observed during recovery, reflects improvements in cellular health and the body’s ability to efficiently utilize nutritional resources. Our findings also highlight that the interaction between caloric and protein intake plays a fundamental role in regulating hydration status. While caloric intake is essential for supporting initial rehydration, protein intake contributes over time to tissue regeneration and intracellular hydration. These hydration markers, therefore, should be integrated into the overall evaluation to ensure a comprehensive and personalized nutritional strategy.

### 4.5. Clinical Implications and Dynamic Nutritional Strategies

The results of this study highlight the need for a nutritional approach based on specific markers to guide and optimize nutritional interventions in patients with AN. The dynamic integration of caloric and protein intake allows for addressing the specific needs of patients at different stages of rehabilitation, adapting nutritional strategies to ensure balanced recovery. Combining traditional anthropometric markers (BW and BMI) with advanced body composition markers (FM, FFM, BCM, and PA) and hydration markers (TBW and ECW) provides a comprehensive and detailed view of recovery progress, enabling the precise calibration of nutritional interventions.

Our findings support and reinforce the concept of dynamic nutritional strategies based on the idea that recovery in AN patients is not a static process but rather a journey requiring continuous adaptations [12,36,37]. Synchronization between caloric and protein intake, monitored through specific markers, helps to balance the recovery of lean and fat mass, promote improved cellular hydration, and support overall metabolic health.

### 4.6. Future Directions

Despite the promising findings of this study, several aspects warrant further investigation. In particular, future research should explore how various factors—such as physical activity, psychological support, and microbiota modulation—interact with nutritional strategies to influence recovery outcomes. Further investigations should also take into account individual variations in treatment responses, with the goal of refining therapeutic protocols and developing more effective and personalized interventions. Advancing this line of research could contribute to a more comprehensive and compassionate approach to the care of patients with AN, ultimately improving long-term recovery and quality of life.

## 5. Conclusions

This study emphasizes the critical role of dynamic and personalized nutritional strategies in the treatment of anorexia nervosa (AN), going beyond traditional measures such as weight and BMI to incorporate advanced markers of body composition, cellular health, and hydration. Recovery in AN is not merely about weight restoration but involves qualitative improvements in metabolic and functional parameters, which are essential for long-term health outcomes. The results underline the multifactorial nature of recovery in AN, where body weight, body composition (e.g., fat mass [FM] and fat-free mass [FFM]), cellular parameters (e.g., body cell mass [BCM] and phase angle [PA]), and hydration status (e.g., total body water [TBW] and extracellular water percentage [ECW%]) must be monitored synergistically. These markers provide a deeper understanding of recovery dynamics, enabling tailored interventions and helping to address imbalances during rehabilitation. In the initial stages of recovery, the careful management of caloric intake is vital to address hypermetabolic states while minimizing excessive fat mass accumulation, which could have psychological and metabolic repercussions. Simultaneously, adequate protein intake supports the preservation of lean tissues and cellular regeneration. Over the medium and long term, the combination of protein adequacy and gradual caloric normalization leads to a more balanced and sustainable recovery, promoting improvements in both body composition and cellular functionality. Hydration markers, particularly BCM, PA, and intracellular water distribution, offer valuable insights into recovery quality, helping to identify areas requiring targeted interventions, such as enhancing intracellular hydration or supporting cellular integrity. The observed reduction in extracellular water percentage and the progressive shift towards intracellular water reflect the effectiveness of integrated strategies in supporting comprehensive recovery. These findings advocate for a paradigm shift in AN treatment, moving from a sole focus on weight gain to a multidimensional approach that considers metabolic, cellular, and functional health. Incorporating advanced markers into clinical practice allows for personalized and adaptive nutritional strategies, ensuring not only weight restoration but also the regeneration of metabolically active tissues and overall physiological improvements.

## Figures and Tables

**Figure 1 nutrients-17-00560-f001:**
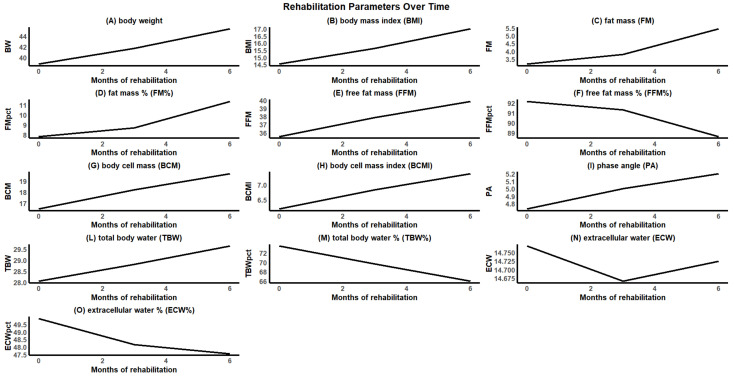
Longitudinal analysis of the trends in body composition and anthropometric parameters measured over the six-month study period: (**A**) body weight (BW); (**B**) body mass index (BMI); (**C**) fat mass (FM); (**D**) fat mass percentage (FM%); (**E**) fat-free mass (FFM); (**F**) fat-free mass percentage (FFM%); (**G**) body cell mass (BCM); (**H**) body cell mass index (BCMI); (**I**) phase angle (PA); (**L**) total body water (TBW); (**M**) total body water percentage (TBW%); (**N**) extracellular water (ECW); (**O**) extracellular water percentage (ECW%).

**Table 1 nutrients-17-00560-t001:** Anthropometric and body composition parameters measured at the start of observations (T0), after 3 months (T1), and 6 months (T2) in AN patients.

	T0	T1	T2	P_(T0 vs. T1)_	P_(T1 vs. T2)_	P_(T0 vs. T2)_
BW (kg)	38.75 ± 6.36	41.68 ± 5.53	45.29 ± 6.54	4.07762 × 10^−14^	1.79095 × 10^−15^	3.01592 × 10^−19^
BMI (kg/m^2^)	14.52 ± 2.12	15.63 ± 1.67	16.99 ± 1.95	7.8825 × 10^−14^	1.41241 × 10^−15^	4.85064 × 10^−19^
FM (kg)	3.17 ± 2.20	3.79 ± 2.67	5.43 ± 3.46	7 × 10^−5^	1.28 × 10^−9^	7.4 × 10^−11^
FM%	7.83 ± 4.09	8.69 ± 5.05	11.37 ± 6.03	0.009913	6.81 × 10^−8^	1.93 × 10^−8^
FFM (kg)	35.53 ± 5.03	37.88 ± 4.01	39.86 ± 4.25	3.79 × 10^−11^	3.61 × 10^−12^	2.22 × 10^−15^
FFM%	92.17 ± 4.09	91.31 ± 5.05	88.63 ± 6.03	0.010481	8.98 × 10^−8^	2.59 × 10^−8^
BCM (kg)	16.48 ± 4.15	18.19 ± 3.18	19.64 ± 3.13	2.84134 × 10^−11^	4.54982 × 10^−10^	4.52918 × 10^−14^
BCMI (kg/m^2^)	6.19 ± 1.53	6.84 ± 1.13	7.38 ± 1.04	3.53 × 10^−11^	2.43 × 10^−10^	3.83 × 10^−14^
PA (°)	4.73 ± 1.04	4.99 ± 0.85	5.20 ± 0.78	0.000167	0.000584	4.95 × 10^−7^
TBW (L)	28.05 ± 3.29	28.81 ± 2.90	29.63 ± 3.16	0.000368	4.89 × 10^−7^	1.33 × 10^−8^
TBW%	73.46 ± 8.58	69.67 ± 6.46	65.96 ± 5.55	3.78 × 10^−8^	1.45 × 10^−11^	6.68 × 10^−14^
ECW (L)	14.77 ± 2.39	14.67 ± 1.88	14.72 ± 1.89	0.593261	0.612387	0.833443
ECW%	49.88 ± 14.78	48.15 ± 13.66	47.56 ± 12.03	0.000169	0.389212	0.007383

Data are expressed as mean ± standard deviations. Abbreviations: BW: body weight; BMI: body mass index; FM: fat mass; FM%: fat mass percentage; FFM: fat-free mass; FFM%: fat-free mass percentage; BCM: body cell mass; BCMI: body cell mass index; PA: phase angle; TBW: total body water; TBW%: total body water percentage; ECW: extracellular water; ECW%: extracellular water percentage.

**Table 2 nutrients-17-00560-t002:** Results of the Generalized Estimating Equations (GEE) analysis examining the relationship between caloric intake, body composition parameters, and time.

**A. BW**	Estimate	Std. Err	Wald	*p* Value	**B. BMI**	Estimate	Std. Err	Wald	*p* Value
Kcal/kg/die	−0.1965	0.0669	8.6381	0.0033	Kcal/kg/die	−0.0815	0.0255	10.2371	0.0014
Time	0.6991	0.5424	1.6612	0.1974	Time	0.1777	0.2007	0.7840	0.3759
Kcal/kg/die time	0.0094	0.0130	0.5214	0.4703	Kcal/kg/die time	0.0057	0.0048	1.4267	0.2323
**C. FM**	Estimate	Std. Err	Wald	*p* Value	**D. FM%**	Estimate	Std. Err	Wald	*p* Value
Kcal/kg/die	−0.0368	0.0251	2.1490	0.1427	Kcal/kg/die	−0.0744	0.0468	2.5254	0.1120
Time	1.2983	0.2506	26.8402	0.0000	Time	2.2488	0.4628	23.6057	0.0000
Kcal/kg/die time	−0.0240	0.0059	16.5107	0.0000	Kcal/kg/die time	−0.0434	0.0111	15.1738	0.0001
**E. FFM**	Estimate	Std. Err	Wald	*p* Value	**F. FFM%**	Estimate	Std. Err	Wald	*p* Value
Kcal/kg/die	−0.1778	0.0531	11.2113	0.0008	Kcal/kg/die	0.0744	0.0468	2.525	0.1120
Time	−0.5159	0.4545	1.2886	0.2563	Time	−2.2488	0.4628	23.6057	0.0000
Kcal/kg/die time	0.0314	0.0130	7.8052	0.0052	Kcal/kg/die time	0.0433	0.0111	15.1738	0.0001
**G. BCM**	Estimate	Std. Err	Wald	*p* Value	**H. BCMI**	Estimate	Std. Err	Wald	*p* Value
Kcal/kg/die	−0.1397	0.0410	11.5918	0.0007	Kcal/kg/die	−0.0521	0.0152	11.6777	0.0006
Time	−0.3857	0.3067	1.5817	0.2085	Time	−0.2219	0.1122	3.9074	0.0481
Kcal/kg/die time	0.0231	0.0077	8.9786	0.0027	Kcal/kg/die time	0.0107	0.0028	14.2799	0.0002
**I. PA**	Estimate	Std. Err	Wald	*p* Value					
Kcal/kg/die	−0.0245	0.0107	5.2481	0.0220					
Time	−0.0926	0.0965	0.9193	0.3377					
Kcal/kg/die time	0.0044	0.0025	3.0758	0.0795					
**L. TBW**	Estimate	Std. Err	Wald	*p* Value	**M. TBW%**	Estimate	Std. Err	Wald	*p* Value
Kcal/kg/die	−0.0069	0.0262	0.0700	0.7913	Kcal/kg/die	0.4527	0.0994	20.7402	0.0000
Time	0.5205	0.2796	3.4659	0.0626	Time	1.4780	0.7523	3.8596	0.0495
Kcal/kg/die time	−0.0067	0.0067	0.9590	0.3274	Kcal/kg/die time	−0.0691	0.0197	12.3327	0.0005
**N. ECW**	Estimate	Std. Err	Wald	*p* Value	**O. ECW%**	Estimate	Std. Err	Wald	*p* Value
Kcal/kg/die	0.0477	0.0233	4.1900	0.0407	Kcal/kg/die	0.1170	0.0785	2.2238	0.1359
Time	0.5790	0.2440	5.6303	0.0177	Time	0.8302	0.7743	1.1498	0.2836
Kcal/kg/die time	−0.0150	0.0063	5.6871	0.0171	Kcal/kg/die time	−0.0311	0.0217	2.0581	0.1514

Abbreviations: BW: body weight; BMI: body mass index; FM: fat mass; FM%: fat mass percentage; FFM: fat-free mass; FFM%: fat-free mass percentage; BCM: body cell mass; BCMI: body cell mass index; PA: phase angle; TBW: total body water; TBW%: total body water percentage; ECW: extracellular water; ECW%: extracellular water percentage.

**Table 3 nutrients-17-00560-t003:** Results of the Generalized Estimating Equations (GEE) analysis examining the relationship between protein intake, body composition parameters, and time.

**A. BW**	Estimate	Std. Err	Wald	*p* Value	**B. BMI**	Estimate	Std. Err	Wald	*p* Value
P g/kg/die	−28.3010	6.3084	20.1263	0.0000	P g/kg/die	−10.0500	2.0832	23.2738	0.0000
Time	−5.1566	1.2878	16.0336	0.0001	Time	−1.9178	0.4453	18.5469	0.0000
P g/kg/die time	3.8241	1.4618	6.8435	0.0089	P g/kg/die time	1.4456	0.5085	8.0822	0.0045
**C. FM**	Estimate	Std. Err	Wald	*p* Value	**D. FM%**	Estimate	Std. Err	Wald	*p* Value
P g/kg/die	−33.4522	10.252	10.6456	0.0011	P g/kg/die	−17.2348	9.9717	2.9872	0.0839
Time	−4.0601	1.2785	10.0854	0.0015	Time	−0.0111	1.1361	0.0001	0.9922
P g/kg/die time	3.0709	1.5625	3.8628	0.0494	P g/kg/die time	−0.6905	1.4186	0.2369	0.6265
**E. FFM**	Estimate	Std. Err	Wald	*p* Value	**F. FFM%**	Estimate	Std. Err	Wald	*p* Value
P g/kg/die	2.9994	4.9697	0.3643	0.5461	P g/kg/die	17.2348	9.9717	2.9872	0.0839
Time	−1.0567	0.7152	2.1828	0.1396	Time	0.0111	1.1361	0.0001	0.9922
P g/kg/die time	0.7534	0.7518	1.0041	0.3163	P g/kg/die time	0.6905	1.4186	0.2369	0.6265
**G. BCM**	Estimate	Std. Err	Wald	*p* Value	**H. BCMI**	Estimate	Std. Err	Wald	*p* Value
P g/kg/die	−2.6712	4.0408	0.4370	0.5086	P g/kg/die	−1.4882	1.3839	1.1563	0.2822
Time	−0.2763	0.3362	0.6753	0.4112	Time	−0.0905	0.1282	0.4983	0.4803
P g/kg/die time	0.1409	0.3365	0.1754	0.6754	P g/kg/die time	0.0582	0.1298	0.2010	0.6539
**I. PA**	Estimate	Std. Err	Wald	*p* Value					
P g/kg/die	−0.7505	0.7923	0.8972	0.3435					
Time	0.1173	0.0925	1.6102	0.2045					
P g/kg/die time	−0.1264	0.0988	1.6361	0.2009					
**L. TBW**	Estimate	Std. Err	Wald	*p* Value	**M. TBW%**	Estimate	Std. Err	Wald	*p* Value
P g/kg/die	1.2879	3.6085	0.1274	0.7212	P g/kg/die	12.9758	7.5653	2.9418	0.0863
Time	−1.5337	1.0626	2.0835	0.1489	Time	−0.5539	0.9274	0.3567	0.5503
P g/kg/die time	1.3217	1.0620	1.5488	0.2133	P g/kg/die time	1.0637	1.1026	0.9307	0.3347
**N. ECW**	Estimate	Std. Err	Wald	*p* Value	**O. ECW%**	Estimate	Std. Err	Wald	*p* Value
P g/kg/die	1.0656	2.3143	0.2119	0.6452	P g/kg/die	3.0020	3.0696	0.9564	0.3281
Time	−0.8824	0.5636	2.4511	0.1174	Time	−0.5166	0.4095	1.5915	0.2071
P g/kg/die time	0.8135	0.5909	1.8947	0.1687	P g/kg/die time	0.5363	0.4391	1.4913	0.2220

Abbreviations: BW: body weight; BMI: body mass index; FM: fat mass; FM%: fat mass percentage; FFM: fat-free mass; FFM%: fat-free mass percentage; BCM: body cell mass; BCMI: body cell mass index; PA: phase angle; TBW: total body water; TBW%: total body water percentage; ECW: extracellular water; ECW%: extracellular water percentage.

## Data Availability

The raw data supporting the conclusions of this article will be made available by the authors on request.

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
