# Peer review of "Dynamic Nutrition Strategies for Anorexia Nervosa: Marker-Based Integration of Calories and Proteins"

_nutrients, 2025, doi:10.3390/nu17030560_

Round 1

Reviewer 1 Report

Comments and Suggestions for Authors

This study aims to 1) Evaluate the interaction between calorie/protein intake and time on quantitative indicators (weight, BMI) and qualitative indicators (BCM, PA, FM%, etc.). 2) Predict recovery pathways and identify indicators to guide personalized nutritional interventions. 3) Optimize rehabilitation strategies and provide evidence-based recommendations to improve clinical outcomes. The study, conducted on medical records of 79 hospitalized patients, concludes that leveraging advanced body composition and cellular indicators alongside weight and BMI can optimize recovery processes and improve long-term health outcomes. While the findings are highly intriguing and appear valuable for developing nutritional treatment strategies for AN (anorexia nervosa), several concerns arise. In order to improve their manuscript, following suggestions were made:

  1. Depression or depressive symptoms often accompany AN. Were the 79 participants receiving any pharmacological treatments during the study?
  2. The study primarily focuses on physical interventions. However, as noted in the DSM-5 criteria, the essence of AN lies in the "intense fear of weight gain or becoming fat" and "disturbance in self-perception of body weight or shape." This study lacks psychological evaluation. If the underlying psychological issues are not addressed, weight loss may recur after discharge, despite improved nutritional status. Were psychological evaluations (e.g., EDE-Q) or therapeutic interventions (e.g., CBT or IPT) conducted during hospitalization?
  3. Did refeeding syndrome occur in any cases?
  4. The study does not present blood data. Were parameters such as albumin, ALT, or AST measured?
  5. The description of the nutritional plan in this study lacks specificity. The statement, “Based on the collected information, a personalized nutritional plan was developed. This plan was periodically adjusted during the hospitalization to account for weight changes and psychophysical progress,” is insufficient. Details on calorie and protein intake timing, distribution, and types of meals are missing. Please provide a detailed description of the hospital’s nutritional program.

Author Response

We sincerely thank the reviewer for taking the time to review this manuscript. Please find our detailed responses below, along with the corresponding revisions and corrections highlighted in track changes in the re-submitted files.

Comments 1: Depression or depressive symptoms often accompany AN. Were the 79 participants receiving any pharmacological treatments during the study? 
Response 1: We appreciate the reviewer’s insightful comment regarding the potential impact of pharmacological treatments on our findings. We acknowledge that depression and depressive symptoms frequently co-occur with anorexia nervosa (AN) and that pharmacological interventions could influence the outcomes. To clarify this aspect, we provide the following details. Each patient enrolled in the study, as well as every inpatient at Villa Miralago, was managed from the moment of admission by a multidisciplinary team. This team included psychiatrists, internal medicine physicians, clinical nutritionists, nurses, psychologists, psychotherapists, educators, dietitians, nutritionists, art therapists, and kinesiologists. Patients received specific pharmacological therapies based on the recommendations of the facility's clinicians and psychiatrists. The pharmacological treatment prescribed following the initial psychiatric evaluation was subsequently monitored and adjusted according to the evolution of the patient’s clinical and psychological condition throughout the hospitalization period. (page 3: paragraph 2,2,1; line 134 – 146)

Comments 2: The study primarily focuses on physical interventions. However, as noted in the DSM-5 criteria, the essence of AN lies in the "intense fear of weight gain or becoming fat" and "disturbance in self-perception of body weight or shape." This study lacks psychological evaluation. If the underlying psychological issues are not addressed, weight loss may recur after discharge, despite improved nutritional status. Were psychological evaluations (e.g., EDE-Q) or therapeutic interventions (e.g., CBT or IPT) conducted during hospitalization?
Response 2: As mentioned in point 1, during their stay at VM, patients undergo psychological and psychiatric assessments, followed by highly individualized therapeutic interventions for managing psychiatric comorbidities and pathological dependencies. As part of these evaluations, standardized psychometric tests are administered, including the EDI-3 (Eating Disorder Inventory-3) for analyzing psychological traits associated with eating disorders, the BUT (Body Uneasiness Test) for assessing body discomfort and dysmorphophobia, and the MMPI-2 (Minnesota Multiphasic Personality Inventory-2) for an in-depth examination of personality profiles and potential associated psychopathologies. These tools allow for a more precise definition of the clinical picture and guide the planning of therapeutic interventions. This part is described in page 3: paragraph 2,2,1; specifically in lines 137 – 143 of the new version of the manuscript.
The treatment model at Villa Miralago is integrated, multidisciplinary, and analytically oriented. • Integrated, because it combines rehabilitative interventions that link nutritional implications to cognitive, relational, emotional, and metabolic aspects. • Multidisciplinary, involving various specialists (psychiatrists, child neuropsychiatrists, internists, psychologists, dietitians, educators, art therapists, kinesiologists, nurses, and healthcare assistants), in line with the best international practices. • Analytically oriented, as it aims for deep changes beyond symptomatic remission, promoting positive behaviors and autonomous health management.
This approach is a direct consequence of the nature of the patients admitted to Villa Miralago, which also hosts individuals with particularly severe eating and nutrition disorders, often characterized by a severe psychopathological condition, requiring a high level of medical and psychiatric support. For this reason, attention to psychiatric aspects is an integral part of the therapeutic process, in a multidisciplinary and highly personalized approach. However, we wish to emphasize that this manuscript focuses exclusively on the physiological and nutritional aspects, while the psychological and psychiatric aspects, due to their complexity and relevance, are the subject of a separate study on the same subjects. Given the breadth of the topic, we deemed it appropriate to develop two distinct analyses: one dedicated to organic recovery, presented in this work, and another focused on the psychological and psychiatric aspects, which will be explored separately.
However, we have taken care to integrate further details  in the Materials and Methods section (as stated above) regarding the aspects related to psychological and psychiatric evaluation and support.

Comments 3: Did refeeding syndrome occur in any cases?
Response 3: In accordance with the reviewer, we recognize the importance of preventing refeeding syndrome (RS), a well-recognized clinical complication that can occur during the early days of refeeding in severely malnourished patients. For this reason, our therapeutic approach strictly follows the protocols recommended by international and national guidelines. The safety and well-being of patients are our top priority, and the practices implemented at Villa Miralago are designed to minimize the risk of refeeding syndrome by continuously monitoring clinical parameters and adapting the nutritional intervention to the individual needs of each patient. The protocol adopted at Villa Miralago involves a detailed medical assessment of the individual patient's risk before initiating refeeding. Caloric intake is started at baseline levels and gradually increased. Sodium intake is limited, and thiamine is provided as a supplement, with close monitoring of electrolytes to intervene promptly in case of imbalances. In patients at moderate risk of RS, caloric intake starts at 20 kcal/kg/day, with increases of 10-20% every 2-3 days until reaching 1800-2200 kcal/day or more, if necessary. For high-risk patients, the approach is more cautious, starting with 5-10 kcal/kg/day and gradually increasing until reaching 20 kcal/kg/day. The nutritional plan is adapted based on the clinical condition of each patient, ensuring constant monitoring to prevent complications. In our study, we strictly adhered to the recommendations, and none of the patients participating in the study experienced this complication.

Comments 4: The study does not present blood data. Were parameters such as albumin, ALT, or AST measured?
Response 4: In accordance with the reviewer, we recognize the importance of blood parameters in the clinical monitoring of patients with anorexia nervosa. Indeed, during the initial assessment of the patient upon admission to Villa Miralago, and in agreement with the reference team, the internist prescribes the necessary blood tests to monitor the patients' rehabilitation, with particular attention to the early stages of the process, when their conditions are more severe. The blood indices used to assess the severity of malnutrition and monitor nutritional and metabolic status include: hemoglobin, hematocrit, white blood cells, total lymphocytes, transferrin, blood glucose, creatinine, vitamin B12, folate, ferritin, glucose, creatinine, ALT and AST transaminases, sodium, potassium, chloride, magnesium, calcium, inorganic phosphate, total proteins, albumin, vitamin D, and zinc. Initially, we had not included this information because our goal was not the analysis of biochemical or hematological markers but the identification of parameters directly related to body composition, which are easily monitored over time and non-invasive. Now we have integrated additional details in the Materials and Methods section, page 3; paragraph 2.2.1; lines 124 – 133.

Comments 5: The description of the nutritional plan in this study lacks specificity. The statement, “Based on the collected information, a personalized nutritional plan was developed. This plan was periodically adjusted during the hospitalization to account for weight changes and psychophysical progress,” is insufficient. Details on calorie and protein intake timing, distribution, and types of meals are missing. Please provide a detailed description of the hospital’s nutritional program.
Response 5: In accordance with reviewer suggestion, we recognize the importance of providing a more specific description of the nutritional program, and we have now included information about the nutritional plane, as well as the distribution and type of meals provided during the patients' stay (page 4; paragraph 2.2.1; line 158- 178). However, it should be noted that these meal plans are dynamically adjusted over time. The multidisciplinary team continuously tailors dietary interventions according to the patient’s evolving clinical status, nutritional requirements, and tolerance, ensuring a fully personalized and responsive nutritional rehabilitation approach.

Reviewer 2 Report

Comments and Suggestions for Authors

I would like to congratulate the authors on their work in the present research. I find the approach noteworthy, as, while not particularly novel, the robustness with which the current results have been presented will, in my opinion, help in designing better recovery plans for this unfortunately prevalent condition. I believe the results adequately address the objectives, and the discussion is well-constructed, adding value to the current literature.

However, in an effort to further enhance the already high quality of the manuscript, I believe it could benefit from the implementation of the following two minor revisions.

Firstly: The exclusion criteria, as far as I understand, need to be reviewed. We must remember that exclusion criteria are not the opposite of inclusion criteria. Rather, they refer, in this case, to those patients who, despite meeting the inclusion criteria, will be excluded for "X" reason. This is addressed in lines 107 to 109 of the current version of your manuscript.

Secondly: In my opinion, the future lines of research, currently included in lines "562 to 567" of the conclusions, could be moved to a section separate from the conclusions.

I hope these "subtle contributions" will further enhance the already commendable quality of the manuscript. I wish you great success in the scientific context of your work and trust that this manuscript will achieve significant bibliometric impact.

Author Response

We sincerely thank the reviewer for taking the time to review this manuscript. Please find our detailed responses below, along with the corresponding revisions and corrections highlighted in track changes in the re-submitted files.

Comments 1: The exclusion criteria, as far as I understand, need to be reviewed. We must remember that exclusion criteria are not the opposite of inclusion criteria. Rather, they refer, in this case, to those patients who, despite meeting the inclusion criteria, will be excluded for "X" reason. This is addressed in lines 107 to 109 of the current version of your manuscript.
Response 1: We thank the reviewer for their valuable observation regarding the exclusion criteria. We acknowledge that exclusion criteria should not merely be the inverse of inclusion criteria but should specifically address cases where patients meet the inclusion criteria yet are excluded for particular reasons. In response to this valuable suggestion, we have carefully revised the exclusion criteria to ensure greater clarity and adherence to this principle. The revised section can be found in page 3, paragraph 2.1, lines 110-115 of the current manuscript.

Comments 2: In my opinion, the future lines of research, currently included in lines "562 to 567" of the conclusions, could be moved to a section separate from the conclusions.
Response 2: We appreciate the reviewer’s suggestion regarding the placement of future research perspectives. Following this recommendation, we have moved the corresponding paragraph to a separate section titled 'Future Directions': page 14, paragraph 4.6, lines 588-596 of the revised manuscript. Additionally, we have slightly expanded and refined the original content to provide a more in-depth discussion on future research directions.